# Early-Stage Geopolymerization Process of Metakaolin-Based Geopolymer

**DOI:** 10.3390/ma15176125

**Published:** 2022-09-03

**Authors:** Xiuyu Zhu, Hao Qian, Hongxiao Wu, Quan Zhou, Huiping Feng, Qiang Zeng, Ye Tian, Shengqian Ruan, Yajun Zhang, Shikun Chen, Dongming Yan

**Affiliations:** 1College of Civil Engineering and Architecture, Zhejiang University, Hangzhou 310058, China; 2Engineering Design and Research Institute of Rocket Force, Beijing 100011, China; 3Zhejiang Jiaotou Shengxing Mining Co., Ltd., Shaoxing 312432, China

**Keywords:** early geopolymerization, spatial filling, physicochemical coupling model, expansion, dehydration

## Abstract

The geopolymerization of aluminosilicate materials in alkaline environments is a complex physicochemical process that greatly influences the microstructure and engineering performances. This work aims to reveal the geopolymerization process of metakaolin-based geopolymer (MKG) in the first 5 d. Physicochemical characteristics of different evolution stages are disposed of in chronological order. The evolutions of electrical resistivity, dehydration process, volume deformation, and ionic concentration are comprehensively analyzed. Results show that chemical dissolution produces large dismantled fragments rather than small free monomers. The formation of a solid matrix follows the “spatial filling rule”, which means that gels grow by locking swelling fragments to form a framework, then densely filling residual space. Based on chemical models, early geopolymerization of MKG can be divided into six stages from the physicochemical perspective as dismantling, locking fixation, free filling, limited filling, second dissolution, and local mending. Those findings expand the understanding of the phase evolution of the early geopolymerization process; thus, the microstructure of MKG can be better manipulated, and its engineering performances can be improved.

## 1. Introduction

Geopolymer is a type of environmentally friendly aluminosilicate binder material utilizing low-cost natural minerals or industrial wastes (metakaolin, fly ash, volcanic ash, coal gangue, etc.) [1,2]. Metakaolin is mainly used to produce geopolymers because it is a common industrial mineral that can be obtained in large quantities with homogeneous properties [3]. Furthermore, metakaolin performs better reactivity in geopolymerization [4]. Alkali activation of metakaolin can be performed by a solution of alkali hydroxide or by alkali salt, e.g., alkali silicate. The process comprises the dissolution of a primary aluminosilicate framework from metakaolin followed by condensation of the free silicate and aluminate species to form a three-dimensional structure [5,6]. The outcomes of the metakaolin-based geopolymer (MKG) are closed to the structure of zeolites, but the linkage of tetrahedra is more disordered in the long range [7]. When they are sealed and stored at warm temperatures, metakaolin-based geopolymers with low Si/Al ratios may transform into sodalite, faujasite, gismondine, and Linde-type A (LTA) zeolite [8,9,10].

Generally, the mechanical properties and durability of MKG directly arise from the hardening process [3]. There are several classical geopolymerization models to explain the process. The model proposed by Glukhovsky defines geopolymerization as three stages: (i) destruction–coagulation; (ii) coagulation–condensation; and (iii) condensation–crystallization [11]. This model contains both chemical changes (i.e., crystallization) and physical changes (i.e., coagulation), but their underlying causal connection needs to be explained in more detail. The refined model from P. Duxson contains five stages, including dissolution, speciation equilibrium, gelation, reorganization, polymerization, and hardening [12]. This five-staged model emphasizes the evolution of gels and water. The most detailed model from J. Deventer offers two different evolution processes. In early stages, reactants experience dissolution and oligomerization to form aluminosilicate oligomers. After that, oligomers form a “Zeolitic phase” through nucleation and crystallization or amorphous gels through polymerization and gelation. The gel may transform into a Zeolitic phase under suitable temperatures and pressures [9]. Those reaction models give basic theoretical guidance for analyzing prominent phase evolution and proposing optimization methods for design and maintenance. The guidance would be clearer if a time coordinate was added for each stage. Furthermore, the early geopolymerization process is accompanied by complex physical changes, such as hardening, chemical expansion, dry shrinkage, heat release, and moisture loss [13,14,15,16,17]. These changes are easy to observe and test; thus, they are suitable landmarks of reaction stages. Therefore, the comprehensive analysis of chemical and physical mechanisms is an important approach to breaking through the predicament.

The geopolymerization process of MKG can be influenced by many factors [3]. The reactive stages in geopolymerization can be flexibly adjusted by changing the component and curing parameters. From the reactive speed and degree aspect, the alkalinity of the alkaline activator and the milling of solid reactants both affect the dissolution rates of the aluminosilicate precursor [18,19]. The reaction rate of polymerization can be adjusted by several methods, such as higher curing temperature, nanomaterial addition, and so on [20,21]. In terms of the chemical structure of the product, the difference in the dissolution rate of Si and Al elements causes the partition of Si-rich gels and Al-rich gels, according to the Fourier transform infrared (FTIR) spectroscopic map [22]. The analysis of polymerization degree with NMR demonstrates two disconnected polymerization stages. The first course before dehydration mainly consumes dissociative oligomers or monomers, forming gels with higher polymerization degrees and more saturated central Si atoms linked by four oxygen bridges. The second polymerization course happens after dehydration and forms more unsaturated central Si linked by three oxygen bridges [23]. The products of specific stages are adjustable by adding silica fume, nanometer Al_2_O_3_ seeds, etc., to change the Si/Al ratio [24,25]. Among the factors, the water/binder ratio can delay the reaction between the mineral particles and the activator and prolong the setting time without new products [3]. Therefore, a high water/binder ratio is a key factor in studying the geopolymerization process without affecting the chemical evolution and outcomes.

This study aims to form an integrated physicochemical coupling model with a unified time base. The water/binder ratios in this paper are higher than industrial applications because they can properly prolong reaction stages and help distinguish reaction stages more clearly. Such conclusions will allow more safety and regulation space in industrial applications. For samples with a specific water/binder ratio in this study, the same raw material source and curing temperature ensure the same reaction speed in different test processes [26]. This is the basis for the time-based comparability of six different experiments. The suitable duration to study early polymerization processes is about 5 days because the mechanical strength at 5 d reaches about 80% of the strength at 28 d [3], and the chemical structure shows no further drastic changes [23]. On this basis, several physical phenomena in early geopolymerization are studied. Resistivity detection provides a comprehensive factor that reflects the degree of the chemical reaction and the solid–liquid–gaseous state changes of the cementitious material system [27,28,29]. Beyond that, experiments on volume evolution are carried out for more details of gel filling effects; experiments on setting time and dehydration aim to reveal the formation of initial solid frameworks; experiments on ion concentration change certify different dissolution and polymerization characteristics after dehydration. Thus, early geopolymerization can be analyzed through physicochemical coupling effects, and the classical geopolymerization model can be further refined.

## 2. Materials and Methods

### 2.1. Sample Preparation

Metakaolin-based geopolymer (MKG) samples were prepared by mixing metakaolin and alkaline activator. The industrial Metamax^®^ metakaolin (MK) was obtained from BASF SE (Ludwigshafen, Germany). The metakaolin powder was characterized by X-ray fluorescence analysis (XRF-1800), and its chemical composition can be seen in Table 1. Figure 1 depicts the particle size distributions of MK, and the average diameter of MK powders is 5.91 μm. The alkaline activator was obtained by mixing water glass (WG), sodium hydroxide (NaOH), and extra water. The modulus (M = n(SiO_2_)/n(Na_2_O)) of water glass is 3.28, and its chemical compositions are shown in Table 2. The sodium hydroxide was in flake form (CP, 98% purity). For extra water, deionized water was used in samples for inductively coupled plasma optical emission spectrometer (ICP-OES) detection, and tap water was used in other samples. The above four raw materials (MK, WG, NaOH, and extra water) were mixed together with the designed ratios as follows: n(SiO_2)_/n(Al_2_O_3_) = 4, n(Na_2_O)/n(Al_2_O_3_) = 1, and the water/binder ratio ranged from 0.6 to 0.8. The water/binder ratio is calculated by Equation (1).
Water/binder ratio = (M_1,water glass_ + M_extra water_)/(M_MK_ + M_2,water glass_ +M_NaOH_)(1)
where M_1,water glass_ is the mass of H_2_O in water glass and M_2,water glass_ is the total mass of the remaining two components, SiO_2_ and Na_2_O, in Table 2. The mix proportions of the five test groups are listed in Table 3.

To prepare samples, water and sodium hydroxide were added to the water glass, and the system was stirred until the solids were completely dissolved. The resulting solution was cooled at room temperature. Next, metakaolin was added to the alkaline activator and stirred at 140 r/min for 60 s and then at 285 r/min for 30 s. The samples were denoted according to the water/binder ratio as G-0.6, G-0.65, G-0.7, G-0.75, and G-0.8.

### 2.2. Non-Contact Electrical Resistivity Test

The mixed MKG slurry of 1.6 L was poured into lidded annular molds. The water/binder ratios of slurry were 0.6, 0.7, and 0.8. The slurry formed a secondary coil of an electromagnetic excitement system, and the resistance signals were recorded every 60 s. At the same time, the temperature signals of the sample and environment were recorded at the same frequency. These samples were cured and tested at about 20 ± 2 °C. The resistance value was corrected by sample height through software to output the resistivity value. The resistivity data were adjusted according to the temperature compensation law because heat is released and accumulated gradually during early geopolymerization. The local temperature increase leads to more side effects on the resistivity calculation in the later period than the early period. Therefore, the recorded resistivity was converted into standard resistivity at 20 °C following Formula (2):*L*_R_ = *L*_0_ × [1 + *α*(*t*_R_−*t*_0_)](2)
where *L*_R_ represents the resistivity that was recorded and output at environmental temperature *t*_R_; *L*_0_ represents the standard resistivity at standard temperature *t*_0_, and *t*_0_ is 20 °C. Saturated concrete performs good thermal conductivity, and its temperature correction factor *α* ranges from 0.022 to 0.030/°C. For most cementitious mortars without coarse aggregates, factor *α* ranges from 0.020 to 0.040/°C [30,31,32]. Thus, 0.030/°C is suitably used for early MKG, considering that it is dense and wet even without coarse aggregates.

The resistivity data were recorded every 60 s, although such frequency and precision with two decimal places are not conducive for a slope. Therefore, the resistivity was smoothed by 100 data points in each window, and the first derivative was taken. Then, the first derivative—known as the slope of resistivity—was smoothed by 1000 data points in each window to obtain a clear slope curve.

### 2.3. Chemical Deformation Test

Chemical deformation during the first 24 h was detected according to ASTM C1608-2017. About 125 mL paste was cast in small vials and consolidated by tapping the vial against a vibrating table. Sodium hydroxide solution with pH = 13 was added to the top of the vial to avoid disturbing the paste. Then, the rubber stopper with inserted capillary tube was placed tightly into the vial. Paraffin oil was applied on the liquid level to minimize water evaporation. The graduation and capacity of the capillary tube were 0.01 mL and 2 mL, respectively. The liquid level was periodically recorded every 30 min.

### 2.4. Autogenous Strain Test

Autogenous strain of MKG during the first 8 d was detected according to ASTM C1698-09 (R-2014). Approximately 185 mL volumes of paste were cast into standard corrugated plastic tubes with two tapered-end plugs. The gauge bonded to one end of the plugs automatically recorded the length change each 10 min at 20 °C.

### 2.5. Setting Time Test

The initial and final setting times of cement were tested using a Vicat apparatus according to GB175-2007. The samples of three groups were cured and tested at 20 °C and 99% humidity.

### 2.6. Water Loss Measurement

For a reliable comparison among different water/binder ratios and key time points, the slurry samples included five water/binder ratios. The pastes, each of 40 mL volume, were cast into centrifuge tubes and sealed. Twelve identical tubes were prepared for each water/binder ratio. All tubes were cured at 22 ± 1.5 °C for 90 h. The slurry gradually solidified and a liquid supernatant formed on top of the tube. The liquid was collected with absorbent cotton paper every 30 h, the mass was weighed, and the average mass change of 12 tubes was recorded.

### 2.7. ICP-OES Test

This study adopted OPTIMA 8000 DV (made by PerkinElmer, Waltham, MA, USA) as the measuring instrument. The slurry samples were made with deionized water and cast into cubes with 20 mm sides. The cubes were cured at 22 ± 1.5 °C for 120 h. At 9 h, 30 h, 60 h, 90 h, and 120 h, one cube from each sample was tested for ion concentration in pores with the soaking method. The detailed procedure consisted of grinding the solid cube into powder, weighing 1.000 ± 0.001 g powder, adding it to 300 mL deionized water, and allowing static dissolution to occur for 45 min. At the end of dissolution, the solution was neutralized with HCl (AR purity) to pH = 6.0–7.0, and deionized water was added to each sample to ensure 500 mL liquid in each beaker. Subsequently, the solution was sieved through a needle cartridge filter to obtain clear liquid without suspended solids. The diameter of the filtration pore was 1.2 μm.

## 3. Results

### 3.1. Resistivity Evolution

The detection of non-contact resistivity started from the time point when raw materials were mixed with the alkaline activator. As shown in Figure 2a and Figure 3, the resistivity of all three samples showed a growing trend. The growing trend was consistent with the polymerization and accumulation of the solid phase [28,29]. The slope of resistivity dropped sharply in the first 4 h, then rose quickly in 4–10 h (Figure 2b). After that, the increase slowed down and reached a plateau at about 15–30 h. From about 40 h, the slope gradually decreased. The plateau and decrease in slope correspond to linear and logarithmic increases in resistivity, respectively, demonstrating two distinct modes of polymerization and accumulation.

The change of resistivity is a comprehensive factor of early geopolymerization. During the period of 0–10 h, the most important chemical change was the dissolution of raw materials [23]. Hence, the initial increase before the plateau (about 0–4 h) implies that the products of chemical dissolution were not small or enough to increase conductivity. The products of chemical dissolution may be large, dismantled fragments due to the stronger instability of aluminate tetrahedrons than silicate tetrahedrons [18]. In the period of 10–50 h (about 0.5–2 d), the resistivity showed an approximatively linear increase. The linear trend of resistivity indicates that the saturated polymerization stage was not subject to obvious disturbances of any byproducts, raw material shortage, or spatial restriction. According to nucleation theory, silicate and aluminate monomers or small chain-shaped silicon-aluminum compound ions start to link together and fix a number of polymerization trigger sites [1,5]. Thus, in this stage, the polymerization of the nucleus’ surface layer can proceed freely. After 50 h, resistivity gradually slowed down and exhibited a logarithmic curve. It indicates that the polymerization is restrained in a solid and strong matrix. According to NMR analysis, these two different polymerization patterns are attributed to the lack and unavailability of free monomers due to dehydration process finishing at 3 d [23].

### 3.2. Chemical Deformation and Autogenous Strain

The detection of chemical deformation aims to show the volume change resulting from the polymerization and reorganization of fresh-state aluminosilicate paste [33,34,35]. In the first 4 h, samples show slight shrinkage, which accords with violent dissolution and destruction of raw materials. Li et al. [16,36] observed the same initial chemical shrinkage, but the shrinkage period lasted shorter. It could be caused by several possible reasons, such as particle granularity, reactivity, component proportion, etc. In 4–10 h, the comparison of MKGs with different water/binder ratios reveals an almost coincident increase in the same pace (Figure 4). A similar increase happens to autogenous strain (Figure 5). The mechanism of synchronous expansion in 4–10 h may be rooted in mineralogical properties. Metakaolin with various granularities can absorb different amounts of free water into Al_2_O_3_·2SiO_2_ layers in zones with a higher degree of crystallinity [37]. Thus, the slightly dissolved products can be soaked thoroughly and expand significantly, and this may be the main reason for synchronous expansion in 4–10 h (i.e., absorption expansion). At the same time, water consumption caused by absorption and dissolution can weaken ionic migration. Expanded products can also block the migration paths of ions and show the same effect. Weak ionic migration can exactly explain the plateau of resistivity in 4–10 h.

In 10–50 h, samples keep expanding in chemical deformation (Figure 4), which reaches a consensus with Li’s work [16,36]. According to Zhu [23], the expansion develops along with intensive ordered polymerization that consumes more Al elements. Li [36] reached a similar conclusion that chemical expansion is associated with the formation of Al-rich products. On this basis, chemical expansion in 10–50 h verifies that abundant gels rapidly polymerized before dehydration is the primary reason for linear resistivity growth. In terms of autogenous strain, autogenous shrinkage in the study of Portland cement refers to the volume shrinkage caused by consuming pore water after solidification. It contains two parts: (i) chemical shrinkage that is restrained by a rigid skeleton in paste; (ii) the loss of internal water during geopolymerization creates capillary stress [38,39,40,41]. Those conceptions are applied and verified in geopolymers [33,34,42,43,44]. In Figure 5, autogenous expansion gradually slows down and even turns into shrinkage during 0.5–2.25 d (about 12–54 h). Capillary stress can be verified to be the main reason for autogenous shrinkage by comparing the water environment of those two deformation experiments. In autogenous strain experiments, pastes are cast and sealed in corrugated plastic tubes, and there is no water supplement. In chemical deformation experiments, pastes are cast and immersed under an alkaline liquid that provides abundant water and pressure. The water supplement and pressure restrain the removal of water from pores and the generation of capillary pressure. Therefore, the abundant gels polymerized in 10–50 h help to form fine pore systems, which means gels are accumulated by inner filling without much restraint considering linear resistivity growth.

After 50 h, chemical expansion keeps still, and autogenous shrinkage gradually accumulates. The study from Li et al. holds that Si-rich gels formed in the later stage of polymerization would cause volume shrinkage after 50 h [16,36]. In this stage, the formation and accumulation of gels cause severe inner pressure and may even change the later formation of new gels. The gel filling effect is accepted to be important to finer pores and denser matrices during long curing periods [34,45]. In fact, the filling process shown as autogenous shrinkage starts early. The MKG with a lower water/binder ratio that polymerizes more intensively turns from expansion to shrinkage earlier and accumulates more autogenous shrinkage (Figure 5). This regularity can also prove the existence of an inner filling process.

### 3.3. Setting Time

The initial and final setting times of MKG with different water/binder ratios were measured by referring to cement paste (Figure 3). Thus, the solidification of these samples mainly occurs at 3.5–9.75 h, which is only in the plateau phase of resistivity. The smaller the water/binder ratio, the earlier the initial setting starts. The intervals between the initial and final settings for these three samples are 140 min, 120 min, and 105 min, respectively. Therefore, if the water/binder ratio is larger, MKG needs less time from the initial to final setting.

These development rules about setting time are related to the “dissolution–polymerization” chemical essence of the geopolymerization process. Setting time indicates a specific polymerization extent that the solid matrix shapes as a block and forms its initial strength, although its capability to withstand external pressure is still limited [34,45]. Higher water content indicates lower alkalinity. According to Yang [46], Al is more rapidly dissolved in the alkaline medium to form the Al-rich gel phase at short reaction times. Subsequently, the Al-rich gel provides rapid linkage and coverage of the remnant precursor particles during the induction period. Therefore, if the water/binder ratio is lower, more Al-rich gels are generated quickly to link the remnant of metakaolin and form an initial aluminosilicate framework; thus, the initial setting time will be shorter. For the same reason, MKGs with lower water/binder ratios require less time to reach the strength corresponding to the final setting. However, rapid generation of initial gels can make products cover the surface of remnant precursor particles. Moreover, the samples with a lower water/binder ratio have less water, which will weaken the free movement of ions and retard the bond linkage [47]. Therefore, the duration from the strength of the initial setting to the strength of the final setting is prolonged, and the reduction of the final setting time is not as much as the initial setting time.

According to the setting time, the construction of the initial matrix framework is finished in 10 h from separated nuclei to the solid matrix. Considering the filling effect of gels since 0.5 d, the initial solid matrix formed after the final setting time is a rough framework with abundant interior space. Therefore, the stacking of polymerized products from the nuclei should be cleared. Considering the rapid absorption expansion in 4–10 h, the products are more likely to be locked and fixed together from random directions rather than accumulated regularly from specific directions such as sedimentation. The nuclei polymerize some oligomers to extend themselves until new oxygen bridges build linkages among outer products and adjacent fragments. That means bigger clusters (growing from nuclei) act as “locks” to fix fragments quickly and form the solid matrix framework.

### 3.4. Water Loss

MKG pastes with water/binder ratios of 0.6–0.8 were continuously observed, and significant dehydration phenomena could be observed in 4–50 h (Figure 3). The dehydration time and performance of different samples are similar. Dehydration can be roughly divided into the following three stages. In the period of 4–10 h, condensed water droplets appeared on the top and inner walls of the test tubes, and more droplets condensed in samples with higher water/binder ratios (Figure 6a). In the period of 10–35 h, the amount and size of condensed water droplets remained unaltered, but many water droplets were produced at the surface of the samples. Droplets gradually grew larger and then connected to the liquid circle or liquid layer (Figure 6b). During 35–50 h, the liquid layer gradually became thicker (Figure 6c).

The gravimetric test results of dehydration could give a clearer demonstration of the total dehydration mass and periodical dehydration speed (Figure 7). Because early dehydration occurs for small liquid droplets, and they are difficult to collect and weigh, the interval of two measurements was set as 30 h. MKG samples of different water/binder ratios had a similar water loss in each stage. The water collected within 0–30 h mainly condensed droplets on the inner wall and partly from separated droplets on the sample surface. More than 90% of the cumulative heat release of MKG happened in 0–24 h [48]. Thus, the main reason for dehydration in this stage is heating release.

During 30–60 h, collected water was mainly from separated water droplets on the surface. Considering that the gel filling effect makes pores finer, separated surface droplets come from internal pore spaces through surface exits [19]. For geopolymers with single mineral material, the gel filling effect indicates that water is spatially replaced by gradually formed aluminosilicate gels and squeezed out from the inner matrix when there is abundant water in the reaction system. During 60–90 h, MKG keeps dehydrating slightly, indicating that inner filling and squeezing exist for a long time, although the matrix has become solid and the dehydration phenomena have stopped.

### 3.5. Ionic Concentration

The detection of ICP-OES can provide quantitative information about free ions in the pore fluid (Figure 8). Several preliminary experiments were performed to eliminate the error or interference of sample preparation. The main intervals between the two measurements were kept the same as in the water loss experiment, and a beginning measurement was added at 9 h soon after the final setting. Since there is sodium silicate in the reactant system, it is reasonable to have an initial concentration of free silicate ions, and the initial concentration of free aluminate should be 0 (at 0 h, as seen in Figure 8). Therefore, the quantitative element analysis of Si should be focused on the descent value of the two samples with adjacent testing times.

In the period of 0–9 h, free silicate ions showed an overall consumption trend (Figure 8a). This trend proves that the earliest chemical reaction stage, known as “dissolution”, can not produce enough monomers for polymerization. The importance of initial dissociative silicate for early polymerization is verified so that a higher content of dissociative silicate is effective in shortening setting time and promoting saturated polymerization. In 9–30 h, the consumption of silicate monomers decreased with the rise of the water/binder ratio, while aluminate monomers were not consumed and even accumulated (Figure 8b). The different consumption tendencies of silicate and aluminate during 0–30 h is important evidence of the partition of Si-rich gel and Al-rich gel. From the consumption contrast of silicate and aluminate, MKG with a higher water/binder ratio tends to fix more silicates in saturated polymerization, then more Si-rich gel or crystal will form [49]. During 30–60 h, the concentration of silicate and aluminate ions in MKGs with lower water/binder ratios exhibited a huge increase. For MKGs with higher water/binder ratios, aluminate accumulated, but silicate was consumed. This stage is important to water loss through the pore structure, so the ion concentrations of samples of lower water/binder ratios were more affected.

During 60–90 h, the ionic concentration of different samples remained basically stable. After that, the silicate and aluminate ions in different samples are obviously consumed during 90–120 h (Figure 8c). According to the water loss experiment, the change of water in the pore structure was weak after 60 h, so the consumption indicates that another stage of intensive polymerization happens after an ionic equilibrium period. According to NMR analysis, the polymerization stage after dehydration mainly consumes partly linked silicate tetrahedrons to harden the aluminosilicate matrix [23]. The secondary intensive increase of the mechanical strength of the matrix during the long-term curing period can also verify such a chemical change [50]. In this study, the samples with lower water/binder ratios consume more ions after 90 h. It proves that continuous curing improves the strength more significantly in samples with lower water/binder ratios.

## 4. Discussion

### 4.1. Spatial Filling Rules of Geopolymerization

The combination of resistivity and volume evolution reveals that the main polymerization products accumulate by continuous inner filling. The analysis of setting time and dehydration indicates that the early solid matrix rapidly formed by linkage among fragments. On this basis, the building process of the solid aluminosilicate framework obeys the spatial sequence that can be denoted by the spatial filling rule. The details are discussed as follows.

In 0–4 h, this stage is more likely to be dismantling rather than dissolving. The increase of resistivity at this stage indicates that these remnants are not small and are big enough to improve conductivity. Lamellar metakaolin is preferentially torn apart from aluminate sites [23,51]. Thus, products of initial dissolution are more likely to be large unfree fragments instead of small free monomers dissolved orderly from edge to center and from surface to inner. In this stage, the increase of resistivity originates from a messy stack of large fragments with absorption expansion.

From 4 h to the initial setting time, the main physicochemical features are the nucleation process and the first dissolution equilibrium. Monomers and oligomers are produced from hydroxylation reactions, then immigrated in liquid and consumed by de-hydroxylation reactions. The variance between those two reactions generates the ionic accumulation and solubility equilibrium [22,45]. Meanwhile, the dismantlement and absorption of fragments after enough soaking time promote each other and generate volume expansion.

From initial setting time to final setting time, a major system change occurs as the quick growth of the solid framework. In this process, grown nuclei act as locks by forming plentiful interconnections among large dismantled fragments and generate a loose framework with huge interconnected pores.

After the final setting time, polymerization outside of nuclei and linkage among clusters continuously proceed; thus, the effect shows reinforced frameworks or densified pores. The gel filling effect begins and causes volume replacement of water inside the framework. When pores are fine enough to generate capillary pressure, some water can be squeezed out. Thus, MKGs with lower water/binder ratios react more intensively in the initial polymerization, meaning stronger frameworks and denser pores. Thus, more water is replaced and squeezed from the inner space of the framework. If the curing environment causes severe water loss, the framework is more likely to resist collapse and shrinkage [15,46,52].

### 4.2. Six-Stage Geopolymerization Process

The evolution of the ionic concentration reveals two distinct ion consumption stages. The first stage happens before 60 h, and the other one is after 90 h. There are also two distinct ionic dissolution equilibrium stages before 9 h and at 60–90 h. Those two dissolution equilibriums provide suitable polymerization conditions for subsequent consumption stages, respectively. However, the first one happens in an abundant water environment before solidification, and the second one happens in pore water after solidification and dehydration. Considering the spatial filling rule and two dissolution equilibriums, the physical and chemical changes that occur in the early geopolymerization can be expounded into six stages, as described below (Figure 9).
(1)Precursor dismantling (0–4 h). Aluminosilicate precursors are dispersed into the water environment. Firstly, the aluminate sites in Al_2_O_3_·2SiO_2_ layers quickly dissolve (Figure 10a), causing lamellar metakaolin to tear apart from those sites into fragments [53]. The wildly stacked fragments show chemical shrinkage and block the transmission of oligomers to improve resistivity. The dissolved aluminate monomers that randomly distribute in liquid or lie on the fragments’ surface act as central trigger sites. Much silicate monomers from water glass are consumed for initial polymerization (Figure 10b) [22,34,54].(2)Locking fixation (4–10 h). The dissolution of fragments and the surface polymerization of the nucleus maintain a dissolution balance in the liquid phase. The number and volume of nuclei continue to increase, which act as locks and form linkages and fixations among disorderly stacked fragments. This phenomenon will generate a loose and unstable framework, which manifests as the physical solidification and initial setting strength. More new nuclei drift away and form linkages in larger zones, indicating that the framework spreads out in space; therefore, both chemical expansion and autogenous expansion keep surging. The system at this stage releases heat and a large amount of water vapor (Figure 10c).(3)Free filling (10–35 h). The dissolution equilibrium is disturbed, and most nuclei grow quickly into clusters. New nuclei are continuously formed and connect more fragments, so chemical expansion rapidly accumulates. Linear increase of resistivity indicates that the water environment in the system is connected, and the inner filling of clusters into frameworks is spatially free. The water environment and abundant free ions allow the formation of more regular and saturated aluminosilicate compound ions [55]. The inner liquid is dehydrated when polymerized gel clusters occupy the space in the aluminosilicate framework. Winding pores are formed as the main drainage path to the surface (Figure 10d).(4)Limited filling (35–60 h). Resistivity gradually turns from a linear increase to a logarithmic increase. Considering dehydration on the sample surface, the main restraint of gel filling comes from internal space. The nuclei considerably grew in size such that they were spatially close and affected each other. Polymerization in some small cavities can no longer obtain enough free ions from pore fluid to form ordered and saturated molecular chains [10,23]. The pore structure is refined, and the capillary pressure leads to squeezing dehydration. The uneven shrinkage or collapse may tear apart the product layers wrapping out of precursors and dissolving them again [45]. Therefore, autogenous expansion gradually turns into shrinkage (Figure 10e).(5)Second dissolution equilibrium (60–90 h). During this stage, autogenous shrinkages gradually accumulate as new gels are formed. At the same time, the dissolution caused by product layer breakage gradually weakens, and a new ionic balance is formed in the local pore fluid (Figure 10f).(6)Local mending (90–120 h). At the end of the second dissolution equilibrium, new polymerization activity mainly occurs in the zones near cracks and cavities caused by the shrinkage and collapse. As the pore structure gradually becomes disconnected, the ions of local polymerization come from nearby pores and cavities [23,53]. The cracks and defect zones are sewed up partly with new linkages or filled by new clusters. This process leads to a secondary strength increase in the long curing period [55] (Figure 10g).

## 5. Conclusions

In this study, experiments for resistivity, volume change, solidification, dehydration, and ion concentration evolution of MKG were cooperatively designed, and the physicochemical nature of early geopolymerization is comprehensively analyzed. The main conclusions are shown as follows.

The outcomes of chemical dissolution are mainly stacked aluminosilicate fragments. Aluminate oligomers are accumulated, but silicate oligomers are consumed in pore fluid during the first 30 h, so the initial polymerization is supported by silicate monomers in raw materials. Thus, a higher module of alkali activator helps shorten the setting time and strengthen the initial matrix.

The gel filling effect starts as early as the final setting, especially in MKG with low water content. Based on rapid setting and subsequent self-desiccation, the geopolymerization process is supposed to follow spatially filling rules, i.e., growing nuclei link dissolved fragments into the aluminosilicate framework. Subsequently, nuclei grow into dense gel clusters to fill the spaces in the framework.

From a physicochemical perspective, geopolymerization can be divided into six stages: precursor dismantling, locking fixation, free filling, limited filling, second dissolution equilibrium, and local mending. These stages evolve with the corresponding physical states, such as chemical shrinkage, solidification, evaporating dehydration, linear resistivity increase, surface dehydration, stable pore ion concentration, and the second intensive ion consumption.

This work analyzes the complete process of the early geopolymerization of MKG in detail and proposes a comprehensive physical-chemical coupling model to provide theoretical support for related research.

## Figures and Tables

**Figure 1 materials-15-06125-f001:**
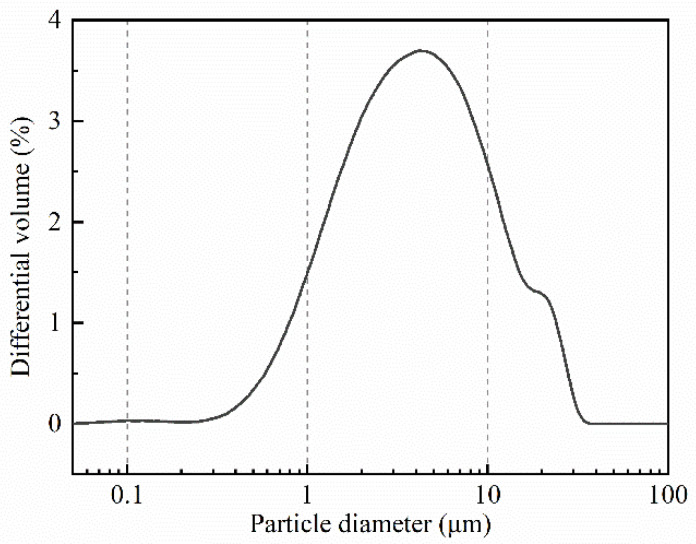
Particle size distribution of metakaolin powders.

**Figure 2 materials-15-06125-f002:**
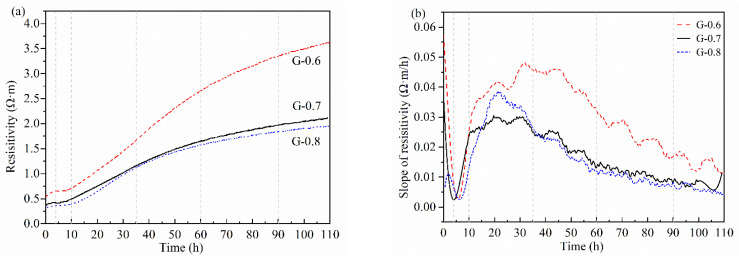
Standard resistivity (**a**) and its slope (**b**) after temperature compensation.

**Figure 3 materials-15-06125-f003:**
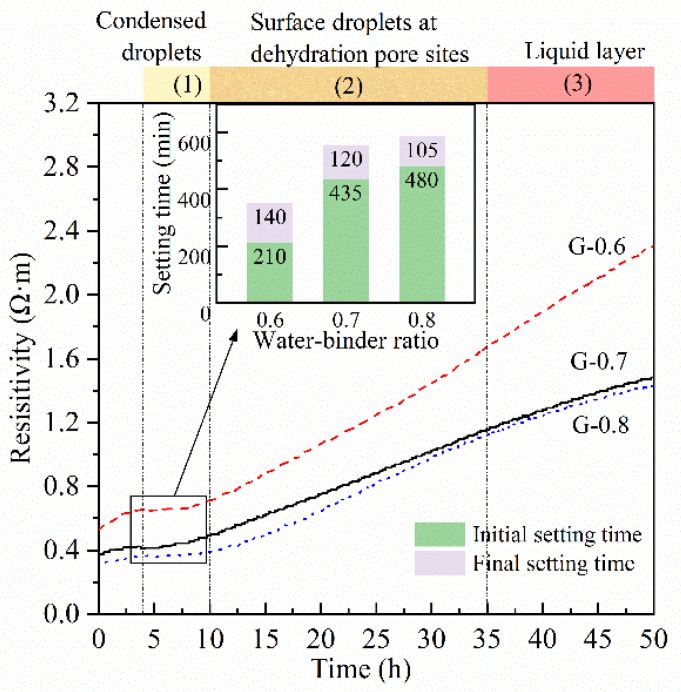
Setting times and dehydration states of samples during 0–50 h.

**Figure 4 materials-15-06125-f004:**
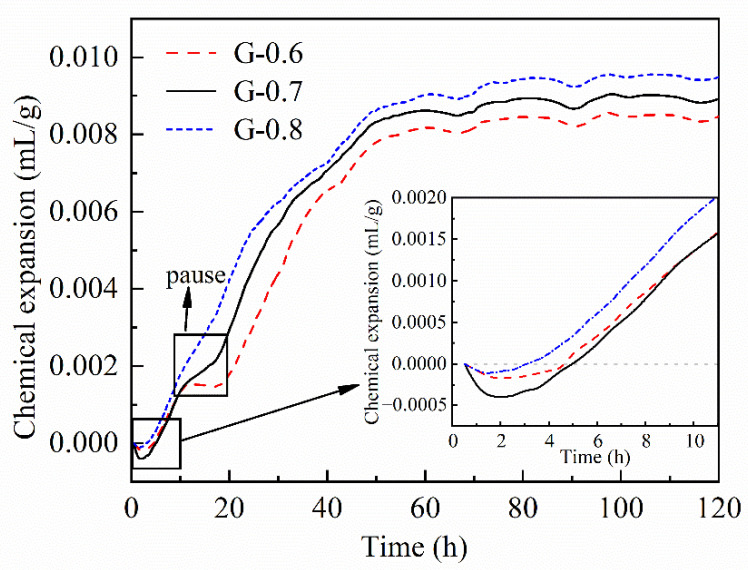
Chemical expansion of G-0.6, G-0.7, and G-0.8 during 0–120 h.

**Figure 5 materials-15-06125-f005:**
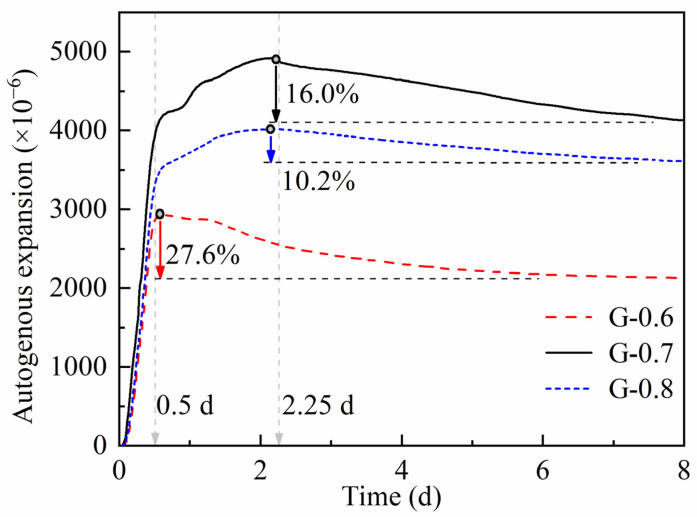
Autogenous expansion of G-0.6, G-0.7, and G-0.8 during 0–8 d.

**Figure 6 materials-15-06125-f006:**
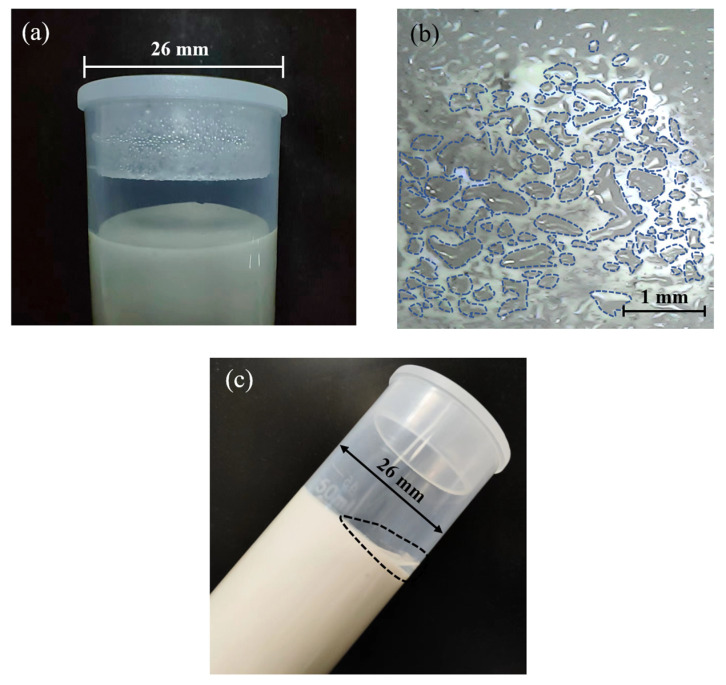
(**a**) Condensed water droplets on the top and inner wall during about 4–10 h. (**b**) Water droplets at the sample surface during about 10–35 h. (**c**) Liquid layer at the surface of sample.

**Figure 7 materials-15-06125-f007:**
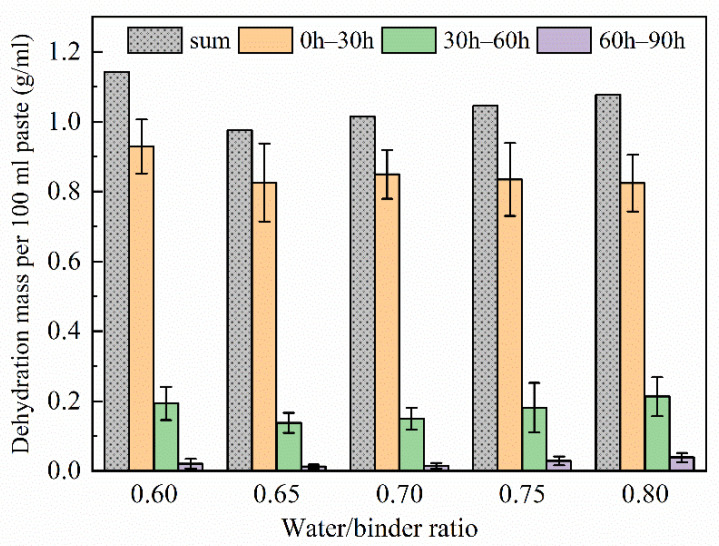
Dehydration speed per 30 h and sum of masses during 0–90 h for five samples.

**Figure 8 materials-15-06125-f008:**
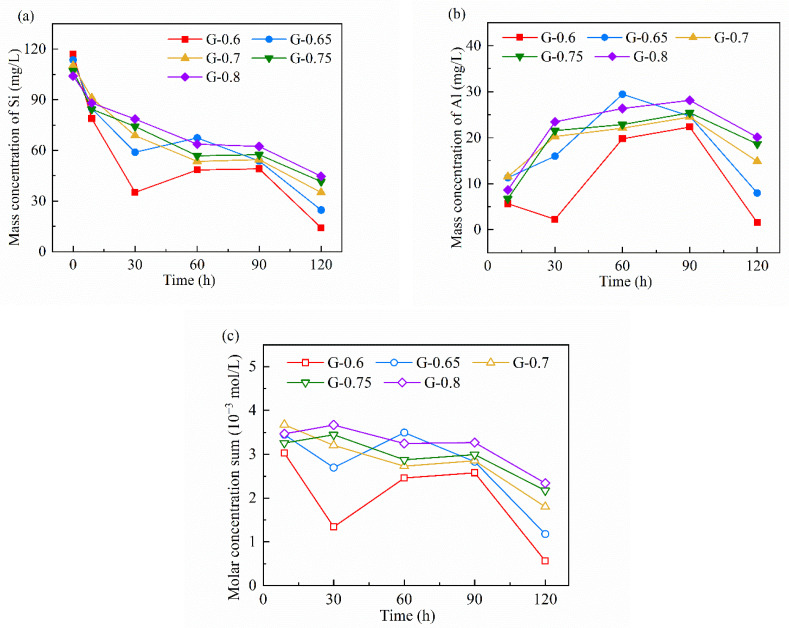
Ionic concentration evolution of Si (**a**) and Al (**b**) by mass concentration and their sum by molar concentration (**c**) during 0–120 h.

**Figure 9 materials-15-06125-f009:**
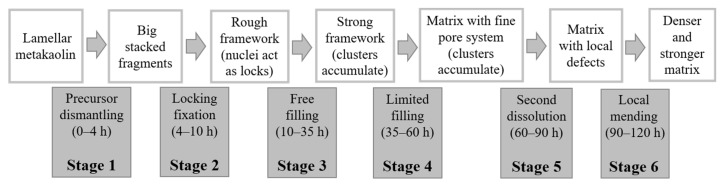
Flow chart of six-stage geopolymerization process.

**Figure 10 materials-15-06125-f010:**
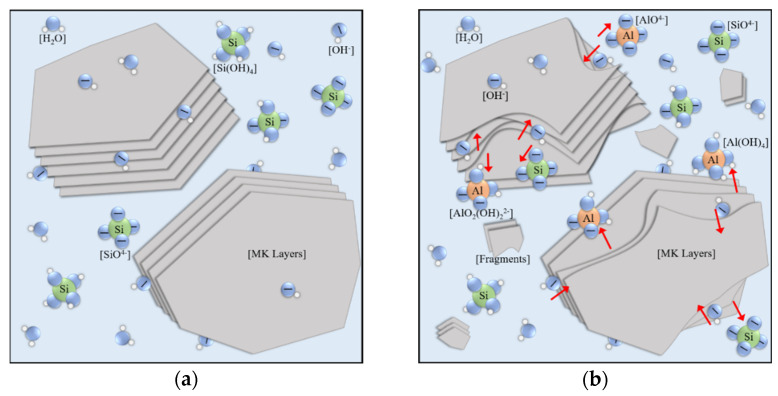
Schematic diagram of early-stage geopolymerization in the six-stage model. (**a**) mixed raw materials; (**b**) precursor dismantling; (**c**) locking fixation; (**d**) free filling; (**e**) limited filling; (**f**) second dissolution equilibrium; (**g**) local mending.

**Table 1 materials-15-06125-t001:** Chemical composition of metakaolin (wt.%).

Component	Al_2_O_3_	SiO_2_	K_2_O	Na_2_O	CaO	TiO_2_	Fe_2_O_3_	LOI
Content	39.68	57.26	0.21	0.27	0.04	1.78	0.43	0.34

**Table 2 materials-15-06125-t002:** Chemical composition of water glass (wt.%).

Component	Na_2_O	SiO_2_	H_2_O
Content	8.2	26.0	65.8

**Table 3 materials-15-06125-t003:** Mix proportions for MKGs (per 100 g).

Test Groups	MK(g)	WG(g)	NaOH(g)	Extra Water (g)	Water/Binder Ratio
G-0.6	38.72	48.25	7.28	5.75	0.6
G-0.65	37.55	46.79	7.06	8.60	0.65
G-0.7	36.45	45.41	6.85	11.29	0.7
G-0.75	35.40	44.12	6.65	13.83	0.75
G-0.8	34.41	42.89	6.47	16.23	0.8

## Data Availability

Not applicable.

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
