# Peer review of "Early-Stage Geopolymerization Process of Metakaolin-Based Geopolymer"

_materials, 2022, doi:10.3390/ma15176125_

Round 1

Reviewer 1 Report

Dear authors,

This study aims to form an integrated physicochemical coupling model with unified time base. The water/binder ratios in this paper are higher than industrial application because they can properly prolong reaction stages and help to distinguish reaction stages more clearly. This work aims at revealing the geopolymerization process of metakaolin-based geopolymer in the first five days. The authors concluded from a physicochemical perspective, geopolymerization can be divided into six stages: precursor dismantling, locking fixation, free filling, limited filling, second dissolution equilibrium, and local mending. These stages evolve with the corresponding physical states, such as chemical shrinkage, solidification and evaporating dehydration, linear resistivity increase, surface dehydration, stable pore ion concentration, and the second intensive ion consumption.

One gets the impression that this investigations will allow more safety and regulation space in industrial application.

My comments:

Lines 102-108….This part of the text originated from Microsoft Word Template??!!This is not acceptable to be found in a paper where serious research is presented. This indicates to me your negligence. Try not to repeat this kind of thing.

Lines 512-620….The list of references should be corrected according Instruction for authors.

Sincerely

Reviewer 2 Report

- Line 101. Section 2 "Materials and Methods". Please check the text of this section and provide information relevant to this section regarding the research reported in the article;
– When describing the characteristics of the MK, it is recommended to indicate the value of its specific surface area;
– Line 109. In section 2.1 “Sample preparation”, please provide a Table with MC-based experimental mixes;
 – Line 119. It should be clarified which "reactant" is meant;
– Line 122. What “remaining two components” do the authors mean when describing the parameter “M2, water glass”;
– Lines 291–292. The sentence "Higher water content indicates higher solubility and lower alkalinity" is controversial. Normally, higher water content leads to lower alkalinity, which, in the case of a geopolymerization process, leads to the lower solubility of aluminosilicate substance. Based on this idea, the statement in the sentence "MKGs with lower water/binder ratios need more time to form solid frameworks" (Lines 299, 300) is also incorrect. Because a lower water/binder ratio needs less time to form solid frameworks Please, revise these sentences.

Round 2

Reviewer 2 Report

Manuscript can be accepted in present form